# Fast Transformers with Clustered Attention

**Apoorv Vyas**[1,2]    **Angelos Katharopoulos**[1,2]    **François Fleuret**[2,3] *
[1]Idiap Research Institute, Switzerland
[2]Ecole Polytechnique Fédérale de Lausanne, Switzerland
[3]University of Geneva, Switzerland
`firstname.lastname@idiap.ch`

## Abstract

Transformers have been proven a successful model for a variety of tasks in sequence modeling. However, computing the attention matrix, which is their key component, has quadratic complexity with respect to the sequence length, thus making them prohibitively expensive for large sequences. To address this, we propose *clustered attention*, which instead of computing the attention for every query, groups queries into clusters and computes attention just for the centroids. To further improve this approximation, we use the computed clusters to identify the keys with the highest attention per query and compute the exact key/query dot products. This results in a model with linear complexity with respect to the sequence length for a fixed number of clusters. We evaluate our approach on two automatic speech recognition datasets and show that our model consistently outperforms vanilla transformers for a given computational budget. Finally, we demonstrate that our model can approximate arbitrarily complex attention distributions with a minimal number of clusters by approximating a pretrained BERT model on GLUE and SQuAD benchmarks with only 25 clusters and no loss in performance.

## 1   Introduction

Sequence modelling is a fundamental task of machine learning, integral in a variety of applications such as neural machine translation [2], image captioning [29], summarization [18], automatic speech recognition [9] and synthesis [19] etc. Transformers [27] have been proven a powerful tool significantly advancing the state-of-the-art for the majority of the aforementioned tasks. In particular, transformers employ self-attention that allows them to handle long sequences without the vanishing-gradient problem inherent in RNNs [13, 1].

Nonetheless, despite their impressive performance, the use of self-attention comes with computational and memory requirements that scale quadratic to the sequence length, limiting their applicability to long sequences. The quadratic complexity becomes apparent if we consider the core mechanism of self-attention, namely splitting the input sequence into queries and keys and then each query attending to all keys. To this end, recently, there has been an increasing interest for developing methods that address this limitation [8, 26, 6, 14].

These methods can be broadly categorized into two distinct lines of work, those that focus on improving the asymptotic complexity of the self-attention computation [6, 15, 14, 24, 3] and those that aim at developing techniques that make transformers applicable to longer sequences without addressing the quadratic complexity of self-attention [8, 26]. The former limits the amount of keys that each query attends to, thus reducing the asymptotic complexity. The latter increases the length of the sequence that a transformer can attend to without altering the underlying complexity of the self-attention mechanism.

In this work, we propose *clustered attention* which is a fast approximation of self-attention. Clustered attention makes use of similarities between queries and groups them in order to reduce the computational cost. In particular, we perform fast clustering using locality-sensitive hashing and K-Means and only compute the attention once per cluster. This results in linear complexity for a fixed number of clusters (§ 3.2). In addition, we showcase that we can further improve the quality of our approximation by separately considering the keys with the highest attention per cluster (§ 3.3). Finally, we provide theoretical bounds of our approximation quality with respect to the full attention (§ 3.2.1, § 3.3.1) and show that our model can be applied for inference of pre-trained transformers with minimal loss in performance.

We evaluate our model on two automatic speech recognition datasets and showcase that clustered attention consistently achieves better performance than vanilla attention when the computational budget is equalized. Moreover, we demonstrate that our proposed attention can approximate a pretrained BERT model on the popular GLUE and SQuAD benchmarks with only 25 clusters and without loss in performance.

## 2    Related Work

In this section, we discuss the most relevant works on scaling transformers to larger sequences. We start by presenting approaches that aim to speed up the attention computation in general. Subsequently, we discuss approaches that speed up transformers without changing the complexity of the attention layer and finally, we summarize the most related works on improving the asymptotic complexity of the attention layer in transformer models.

### 2.1    Attention Improvements Before Transformers

Attention has been an integral component of neural networks for sequence modelling for several years [2, 29, 5]. However, its quadratic complexity with respect to the sequence length hinders its applicability on large sequences.

Among the first attempts to address this was the work of Britz et al. [4] that propose to aggregate the information of the input sequence into fewer vectors and perform attention with these fewer vectors, thus speeding up the attention computation and reducing the memory requirements. However, the input aggregation is performed using a learned but fixed matrix that remains constant for all sequences, hence significantly limiting the expressivity of the model. Similarly, Chiu & Raffel [7] limit the amount of accessible elements to the attention, by attending monotonically from the past to the future. Namely, if timestep $i$ attends to position $j$ then timestep $i + 1$ cannot attend to any of the earlier positions. Note that in order to speed up the attention computation, the above methods are limiting the number of elements that each layer attends to. Recently, some of these approaches have also been applied in the context of transformers [17].

### 2.2    Non-asymptotic Improvements

In this section, we summarize techniques that seek to apply transformers to long sequences without focusing on improving the quadratic complexity of self-attention. The most important are Adaptive Attention Span Transformers [26] and Transformer-XL [8].

Sukhbaatar et al. [26] propose to limit the self-attention context to the closest samples (attention span), in terms of relative distance with respect to the time step, thus reducing both the time and memory requirements of self-attention computation. This is achieved using a masking function with learnable parameters that allows the network to increase the attention span if necessary. Transformer-XL [8], on the other hand, seeks to increase the effective sequence length by introducing segment-level recurrent training, namely splitting the input into segments and attending jointly to the previous and the current segment. The above, combined with a new relative positional encoding results in models that attend to more distant positions than the length of the segment used during training.

Although both approaches have been proven effective, the underlying limitations of self-attention still remains. Attending to an element that is $N$ timesteps away requires $\mathcal{O}\left(N^2\right)$ memory and computation. In contrast, our model trades-off a small error in the computation of the full attention for an improved *linear* asymptotic complexity. This makes processing long sequences possible.

## 2.3 Improvements in Asymptotic Complexity

Child et al. [6] factorize the self-attention mechanism in local and strided attention. The local attention is computed between the $C$ nearest positions and the strided attention is computed between positions that are $C$ steps away from each other. When $C$ is set to $\sqrt{N}$ the total asymptotic complexity becomes $\mathcal{O}\left(N\sqrt{N}\right)$ both in terms of memory and computation time. With the aforementioned factorization, two self-attention layers are required in order for any position to attend to any other position. In addition, the factorization is fixed and data independent. This makes it intuitive for certain signals (e.g. images), however in most cases it is arbitrary. In contrast, our method automatically groups the input queries that are similar without the need for a manually designed factorization. Moreover, in our model, information flows always from every position to every other position.

Set Transformers [15] compute attention between the input sequence $X$, of length $N$ and a set of trainable parameters, $I$, called inducing points to get a new sequence $H$, of length $M << N$. The new sequence $H$ is then used to compute the attention with $X$ to get the output representation. For a fixed $M$, the asymptotic complexity becomes linear with respect to the sequence length. Inducing points are expected to encode some global structure that is task specific. However, this introduces additional model parameters for each attention layer. In contrast to this, we use clustering to project the input to a fixed sequence of smaller length without any increase in the number of parameters. Moreover, we show that not only our method has the same asymptotic complexity, it can also be used to speed up inference of pretrained models without additional training.

Recently, Kitaev et al. [14] introduced Reformer. A method that groups positions based on their similarity using locality-sensitive hashing (LSH) and only computes the attention within groups. For groups of fixed size, the asymptotic complexity of Reformer becomes linear with respect to the sequence length. Note that Reformer constrains the queries and keys of self-attention to be equal. As a result, it cannot be applied to neural machine translation, image captioning or memory networks, or generally any application with heterogenous queries and keys. In addition, as it uses hash collisions to form groups it can only handle a small number of bits, thus significantly reducing the quality of the grouping. Instead, our method uses clustering to group the queries, resulting in significantly better groups compared to hash collisions.

## 3 Scaling Attention with Fast Clustering

In this section, we formalize the proposed method for approximate softmax attention. In § 3.1, we first discuss the attention mechanism in *vanilla transformers* and present its computational complexity. We then introduce *clustered attention* in § 3.2 and show that for queries close in the Euclidean space, the attention difference can be bounded by the distance between the queries. This property allows us to reduce the computational complexity by clustering the queries. Subsequently, in § 3.3 we show that we can further improve the approximation by first extracting the top-$k$ keys with the highest attention per cluster and then computing the attention on these keys separately for each query that belongs to the cluster. A graphical illustration of our method is provided in the supplementary material.

### 3.1 Vanilla Attention

For any sequnce of length $N$, the standard attention mechanism that is used in transformers is the dot product attention introduced by Vaswani et al. [27]. Following standard notation, we define the attention matrix $A \in \mathbb{R}^{N \times N}$ as,

$$A = \text{softmax}\left(\frac{QK^T}{\sqrt{D_k}}\right), \tag{1}$$

where $Q \in \mathbb{R}^{N \times D_k}$ denotes the *queries* and $K \in \mathbb{R}^{N \times D_k}$ denotes the *keys*. Note that $\text{softmax}(\cdot)$ is applied row-wise. Using the attention weights $A$ and the values $V \in \mathbb{R}^{N \times D_v}$, we compute the new values $\hat{V}$ as follows,

$$\hat{V} = AV. \tag{2}$$

An intuitive understanding of the attention, as described above, is that given $Q, K, V$ we create new values $\hat{V}$ as the weighted average of the old ones, where the weights are defined by the attention matrix $A$. Computing equation 1 requires $\mathcal{O}\left(N^2 D_k\right)$ operations and the weighted average of equation 2 requires $\mathcal{O}\left(N^2 D_v\right)$. This results in an asymptotic complexity of $\mathcal{O}\left(N^2 D_k + N^2 D_v\right)$.

## 3.2 Clustered Attention

Instead of computing the attention matrix for all queries, we group them into $C$ clusters and compute the attention only for these clusters. Then, we use the same attention weights for queries that belong to the same cluster. As a result, the attention computation now becomes $\mathcal{O}\left(NCD_k\right)$, where $C \ll N$.

More formally, let us define $S \in \{0, 1\}^{N \times C}$, a partitioning of the queries $Q$ into $C$ non-overlapping clusters, such that, $S_{ij} = 1$, if the $i$-th query $Q_i$ belongs to the $j$-th cluster and $0$ otherwise. Using this partitioning, we can now compute the *clustered attention*. First, we compute the cluster centroids as follows,

$$Q_j^c = \frac{\sum_{i=1}^{N} S_{ij} Q_i}{\sum_{i=1}^{N} S_{ij}}, \tag{3}$$

where $Q_j^c$ is the centroid of the $j$-th cluster. Let us denote $Q^c \in \mathbb{R}^{C \times D_k}$ as the centroid matrix. Now, we can compute the clustered attention as if $Q^c$ were the queries. Namely, we compute the clustered attention matrix $A^c \in \mathbb{R}^{C \times N}$

$$A^c = \text{softmax}\left(\frac{Q^c K^T}{\sqrt{D_k}}\right) \tag{4}$$

and the new values $\hat{V}^c \in \mathbb{R}^{C \times D_v}$

$$\hat{V}^c = A^c V. \tag{5}$$

Finally, the value of the $i$-th query becomes the value of its closest centroid, namely,

$$\hat{V}_i = \sum_{j=1}^{C} S_{ij} \hat{V}_j^c. \tag{6}$$

From the above analysis, it is evident that we only need to compute the attention weights and the weighted average of the values *once per cluster*. Then, we can broadcast the same value to all queries belonging to the same cluster. This allows us to reduce the number of dot products from $N$ for each query to $C$ for each cluster, which results in an asymptotic complexity of $\mathcal{O}\left(NCD_k\right) + \mathcal{O}\left(CND_v\right)$.

Note that in practice, we use multi-head attention, this means that two queries belonging to the same cluster can be clustered differently in another attention head. Moreover, the output of the attention layer involves residual connections. This can cause two queries belonging to the same cluster to have different output representations. The combined effect of residual connections and multi-head attention allows new clustering patterns to emerge in subsequent layers.

### 3.2.1 Quality of the approximation

From the above, we show that grouping queries into clusters can speed-up the self-attention computation. However, in the previous analysis, we do not consider the effects of clustering on the attention weights $A$. To address this, we derive a bound for the approximation error. In particular, we show that the difference in attention can be bounded as a function of the Euclidean distance between the queries.

**Proposition 1.** *Given two queries $Q_i$ and $Q_j$ such that $\|Q_i - Q_j\|_2 \le \epsilon$,*

$$\left\|\text{softmax}\left(Q_i K^T\right) - \text{softmax}\left(Q_j K^T\right)\right\|_2 \le \epsilon \left\|K\right\|_2, \tag{7}$$

*where $\|K\|_2$ denotes the spectral norm of $K$.*

*Proof.* Given that $\text{softmax}\left(\cdot\right)$ has Lipschitz constant less than 1 [10],

$$\begin{aligned}
&\left\|\text{softmax}\left(Q_i K^T\right) - \text{softmax}\left(Q_j K^T\right)\right\|_2 \\
&\le \left\|Q_i K^T - Q_j K^T\right\|_2 \\
&\le \epsilon \left\|K\right\|_2
\end{aligned} \tag{8}$$

$\square$

Proposition 1 shows that queries that are close in Euclidean space have similar attention distributions. As a result, the error in the attention approximation for the $i$-th query assigned to the $j$-th cluster can be bounded by its distance from the cluster centroid $Q_j^c$.

### 3.2.2 Grouping the Queries

From the discussion, we have shown that given a representative set of queries, we can approximate the attention with fewer computations. Thus, now the problem becomes finding this representative set of queries. K-Means clustering minimizes the sum of squared distances between the cluster members, which would be optimal given our analysis from § 3.2.1. However, for a sequence of length $N$ one iteration of Lloyd's algorithm for the K-Means optimization problem has an asymptotic complexity $\mathcal{O}\left(NCD_k\right)$. To speed up the distance computations, we propose to use *Locality-Sensitive Hashing* (LSH) on the queries and then K-Means in Hamming space. In particular, we use the sign of random projections [25] to hash the queries followed by K-Means clustering with hamming distance as the metric. This results in an asymptotic complexity of $\mathcal{O}\left(NCL + CBL + ND_kB\right)$, where $L$ is the number of Lloyd iterations and $B$ is the number of bits used for hashing.

### 3.3 Improving clustered attention

In the previous section, we show that clustered attention provides a fast approximation for softmax attention. In this section, we discuss how this approximation can be further improved by considering separately the keys with the highest attention. To intuitively understand the importance of the above, it suffices to consider a scenario where a key with low attention for some query gets a high attention as approximated with the cluster centroid. This can happen when the number of clusters are too low or due to the convergence failure of K-Means. For the clustered attention, described in § 3.2, this introduces significant error in the computed value. The variation discussed below addresses such limitations.

After having computed the clustered attention $A^c$ from equation 4, we find the $k$ keys with the highest attention for each cluster. The main idea then is to improve the attention approximation on these top-$k$ keys for each query that belongs to the cluster. To do so, we first compute the dot product attention as defined in equation 1 on these top-$k$ keys for all queries belonging to this cluster. For any query, the computed attention on these top-$k$ keys will sum up to one. This means that it cannot be directly used to substitute the clustered-attention on these keys. To address this, before substition, we scale the computed attention by the total probability mass assigned by the clustered attention to these top-$k$ keys.

More formally, we start by introducing $T \in \{0, 1\}^{C \times N}$, where $T_{ji} = 1$ if the $i$-th key is among the top-$k$ keys for the $j$-th cluster and 0 otherwise. We can then compute the probability mass, let it be $\hat{m}_j$, of the top-$k$ keys for the $j$-th cluster, as follows

$$\hat{m}_j = \sum_{i=1}^{N} T_{ji} A_{ji}^c. \tag{9}$$

Now we formulate an improved attention matrix approximation $A^t \in \mathbb{R}^{N \times N}$ as follows

$$A_{il}^t = \begin{cases} \frac{\hat{m}_j \exp\left(Q_i K_l^T\right)}{\sum_{r=1}^{N} T_{jr} \exp(Q_i K_r^T)} & \text{if } T_{jl} = 1 \\ A_{jl}^c & \text{otherwise} \end{cases}. \tag{10}$$

Note that in the above, $i$ denotes the $i$-th query belonging to the $j$-th cluster and $\sqrt{D_k}$ is ommited for clarity. In particular, equation 10 selects the clustered attention of equation 4 for keys that are not among the top-$k$ keys for a given cluster. For the rest, it redistributes the mass $\hat{m}_j$ according to the dot product attention of the queries with the top-$k$ keys. The corresponding new values, $\hat{V} \in \mathbb{R}^{N \times D_v}$, are a simple matrix product of $A^t$ with the values,

$$\hat{V} = A^t V. \tag{11}$$

Equation 11 can be decomposed into clustered attention computation and two sparse dot products, one for every query with the top-$k$ keys and one for the top-$k$ attention weights with the corresponding values. This adds $\mathcal{O}\left(Nk \max\left(D_k, D_v\right)\right)$ to the asymptotic complexity of the attention approximation of equation 4.

### 3.3.1 Quality of the approximation

In the following, we provide proof that improved clustered attention (eq. 10) is a direct improvement over the clustered attention (eq. 4), in terms of the $L_1$ distance from the attention matrix $A$.

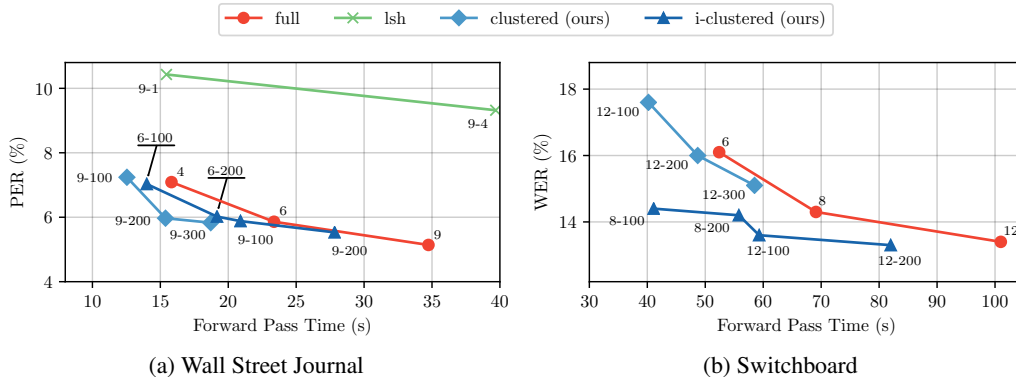

Figure 1: We compare the achieved performance of various transformer models under an equalized computational budget. The numbers near the datapoints denote the number of layers and number of clusters or hashing rounds where applicable. i-clustered is consistently better than all baselines for a given computational budget both in WSJ and Switchboard datasets. The details can be found in § 4.1 and § 4.2 respectively.

**Proposition 2.** *For the $i$-th query belonging to the $j$-th cluster, the improved clustered attention $A_i^t$ and clustered attention $A_j^c$ relate to the full attention $A_i$ as follows,*

$$\left\|A_i^t - A_i\right\|_1 \leq \left\|A_j^c - A_i\right\|_1 \tag{12}$$

Due to lack of space, the proof of the above proposition is presented in the supplementary material. From equation 12 it becomes evident that improved clustered attention will always approximate the full attention better compared to clustered attention.

## 4 Experiments

In this section, we analyze experimentally the performance of our proposed method. Initially, we show that our model outperforms our baselines for a given computational budget on a real-world sequence to sequence task, namely automatic speech recognition on two datasets, the Wall Street Journal dataset (§ 4.1) and the Switchboard dataset (§ 4.2). Subsequently, in § 4.3, we demonstrate that our model can approximate a pretrained BERT model [16] on the GLUE [28] and SQuAD [23] benchmarks with minimal loss in performance even when the number of clusters is less than one tenth of the sequence length. Due to lack of space, we also provide, in the supplementary material, a thorough benchmark that showcases the linear complexity of *clustered attention* and an ablation study regarding how the number of clusters scales with respect to the sequence length.

We compare our model with the vanilla transformers [27], which we refer to as **full** and the Reformer [14], which we refer to as **lsh-X**, where $X$ denotes the rounds of hashing. We refer to *clustered attention*, introduced in § 3.2, as **clustered-X** and to *improved clustered attention*, introduced in § 3.3, as **i-clustered-X**, where $X$ denotes the number of clusters. Unless mentioned otherwise we use $k = 32$ for the top-$k$ keys with improved clustered.

All experiments are conducted using NVidia GTX 1080 Ti with 11GB of memory and all models are implemented in PyTorch [20]. For Reformer we use a PyTorch port of the published code. Note that we do not use reversible layers since it is a technique that could be applied to all methods. Our PyTorch code can be found at `https://clustered-transformers.github.io`.

### 4.1 Evaluation on Wall Street Journal (WSJ)

In our first experiment, we employ the Wall-Street Journal dataset [21]. The input to all transformers is 40-dimensional filter-bank features with fixed positional embeddings. We train using Connectionist Temporal Classification (CTC) [12] loss with phonemes as ground-truth labels. The approximate average and maximum sequence lengths for the training inputs are 780 and 2500 respectively.

**Speed Accuracy Trade-off:** We start by comparing the performance of our proposed model with various transformer variants under an equalized computational budget. To this end, we train *full* with 4, 6 and 9 layers to get a range of the required computation time and achieved *phone error rate* (PER). Similarly, we train *i-clustered* with 6 and 9 layers. Both models are trained with 100 and 200 clusters. We also train *clustered* with 9 layers, and 100, 200 and 300 clusters. Finally, we train Reformer with 9 layers, and 1 and 4 hashing rounds. We refer the reader to our supplementary for the specifics of all transformer architectures as well as their training details. In figure 1a, we plot the achieved PER on the validation set with respect to the required time to perform a full forward pass. Our *i-clustered* achieves lower PER than all other baselines for a given computational budget.

**Approximation Quality:** To assess the approximation capabilities of our method, we train different transformer variants on the aforementioned task and evaluate them using other self-attention implementations during inference. As the Reformer requires the queries to be identical to the keys to evaluate its approximation ability we also train a full attention model with shared queries and keys, which we refer to as **shared-full**. Note that both clustered attention and improved clustered attention can be used for approximating shared-full, simply by setting keys to be equal to queries. Table 1 summarizes the results. We observe that improved clustered attention (7-8 rows) achieves the lowest phone error rate in every comparison. This implies that it is the best choice for approximating pre-trained models. In addition, we also note that as the number of clusters increases, the approximation improves as well. Furthermore, to show that the top keys alone are not sufficient for

|  |  | Train with | | | | | |
|---|---|---|---|---|---|---|---|
|  |  | full | shared-full | lsh-1 | lsh-4 | clustered-100 | i-clustered-100 |
| | full | 5.14 | - | - | - | 7.10 | 5.56 |
| | shared-full | - | 6.57 | 25.16 | 41.61 | - | - |
| | lsh-1 | - | 71.40 | 10.43 | 13.76 | - | - |
| **Evaluate with** | lsh-4 | - | 64.29 | 9.35 | 9.33 | - | - |
| | clustered-100 | 44.88 | 40.86 | 68.06 | 66.43 | 7.06 | 18.83 |
| | clustered-200 | 21.76 | 25.86 | 57.75 | 57.24 | 6.34 | 8.95 |
| | i-clustered-100 | 9.29 | 13.22 | 41.65 | 48.20 | 8.80 | 5.95 |
| | i-clustered-200 | 6.38 | 8.43 | 30.09 | 42.43 | 7.71 | 5.60 |
| | oracle-top | 17.16 | 77.18 | 43.35 | 59.38 | 24.32 | 6.96 |

Table 1: We report validation phone error rate (PER) on the WSJ dataset (§ 4.1). We train with one model and evaluate with another to assess the approximation abilities of different models. Underline denotes training and testing with the same model. Improved cluster (rows 7-8) approximates the full and the shared-full significantly better than all the other fast attention methods.

approximating *full*, we also compare with an attention variant, that for each query only keeps the 32 keys with the highest attention. We refer to the latter as **oracle-top**. We observe that oracle-top achieves significantly larger phone error rate than improved clustered in all cases. This implies that improved clustered attention also captures the significant long tail of the attention distribution.

**Convergence Behaviour:** In Table 2, we report the required time per epoch as well as the total training time for all transformer variants with 9 layers. For completeness, we also provide the corresponding phone error rates on the test set. We observe that clustered attention is more than two times faster than full (per epoch) and achieves significantly lower PER than both Reformer variants (lsh-1 and lsh-4). Improved clustered is the only method that is not only faster per epoch but also in total wall-clock time required to converge.

## 4.2 Evaluation on Switchboard

We also evaluate our model on the Switchboard dataset [11], which is a collection of $2,400$ telephone conversations on common topics among $543$ strangers. All transformers are trained with lattice-free MMI loss [22] and as inputs we use $80$-dimensional filter-bank features with fixed positional embeddings. The average input sequence length is roughly $534$ and the maximum sequence length is approximately 3850. Details regarding the transformer architectures as well as their training details are provided in the supplementary.

|  | full | lsh-1 | lsh-4 | clustered-100 | i-clustered-100 |
|---|---|---|---|---|---|
| PER (%) | 5.03 | 9.43 | 8.59 | 7.50 | 5.61 |
| Time/Epoch (s) | 2514 | 1004 | 2320 | 803 | 1325 |
| Convergence Time (h) | 87.99 | 189.64 | 210.09 | 102.15 | 72.14 |

Table 2: We report the test PER, the time per training epoch (in seconds) and the wall-clock time required for the convergence of each model (in hours).

**Speed Accuracy Trade-off:** Similar to § 4.1, we compare the performance of various transformer models given a specific computational budget. To this end, we train *full* with $6, 8$ and $12$ layers. Similarly, we train *i-clustered* with $8$ and $12$ layers; both with $100$ and $200$ clusters. Finally, we also train *clustered* with $12$ layers, and $100, 200$ and $300$ clusters. In figure 1b, we plot the achieved word error rate (WER) in the validation set of Switchboard with respect to the required time to perform a full forward pass. Our *i-clustered* is consistently better than *full* for a given computational budget. In particular, for a budget of approximately $50$ seconds, improved clustered achieves more than $2$ percentage points lower WER. Furthermore, we note that it is consistently better than clustered attention for all computational budgets.

**Convergence Behaviour:** Table 3 summarizes the computational cost of training the transformer models with $12$ layers in the Switchboard dataset as well as the WER in the test set. We observe that due to the larger sequences in this dataset both clustered and i-clustered are faster to train per epoch and with respect to total required wall-clock time.

|  | full | clustered-100 | i-clustered-100 |
|---|---|---|---|
| WER (%) | 15.0 | 18.5 | 15.5 |
| Time/Epoch (h) | 3.84 | 1.91 | 2.57 |
| Convergence Time (h) | 228.05 | 132.13 | 127.44 |

Table 3: We report the test set WER, the time per training epoch (in hours) and the wall-clock time required for the convergence of each model (in hours).

### 4.3 RoBERTa Approximation

To highlight the ability of our model to approximate arbitrarily complicated attention distributions, we evaluate our proposed method on the approximation of a fine-tuned RoBERTa model [16] on the GLUE [28] and SQuAD [23] benchmarks. In particular, we evaluate on $10$ different tasks, among which there are tasks such as question answering (SQuAD) and textual entailment (RTE), which exhibit arbitrary and sparse attention patterns.

For the GLUE tasks, the maximum sequence length is $128$ while for SQuAD, it is $384$. For each task, we use $25$ clusters for approximation which is less than $20\%$ and $10\%$ of the input sequence length for GLUE and SQuAD tasks respectively. In Table 4, we summarize the performance per task. We observe that improved clustered performs as well as the full transformer in all tasks but SQuAD, in which it is only marginally worse. Moreover, we note that clustered performs significantly worse in tasks that require more complicated attention patterns such as SQuAD and RTE. For inference time, *full* was faster than the *clustered* attention variants due to short sequence lengths.

|  | CoLA | MNLI | MRPC | QNLI | QQP | RTE | SST-2 | STS-B | WNLI | SQuAD |
|---|---|---|---|---|---|---|---|---|---|---|
| full | 0.601 | 0.880 | 0.868 | 0.929 | 0.915 | 0.682 | 0.947 | 0.900 | 0.437 | 0.904 |
| clustered-25 | 0.598 | 0.794 | 0.436 | 0.746 | 0.894 | 0.498 | 0.944 | 0.789 | 0.437 | 0.006 |
| i-clustered-25 | 0.601 | 0.880 | 0.873 | 0.930 | 0.915 | 0.704 | 0.947 | 0.900 | 0.437 | 0.876 |

Table 4: We report the performance on GLUE and SQuAD benchmarks. Following common practice, we report accuracy for all tasks except STS-B and SQuAD, where we report Pearson correlation and F1-score respectively. For all metrics higher is better.

# 5 Conclusions

We have presented *clustered attention* a method that approximates vanilla transformers with significantly lower computational requirements. In particular, we have shown that our model can be up to $2\times$ faster during training and inference with minimal loss in performance. In contrast to recent fast variations of transformers, we have also shown that our method can efficiently approximate pre-trained models with full attention while retaining the linear asymptotic complexity.

The proposed method opens several research directions towards applying transformers on long sequence tasks such as music generation, scene flow estimation etc. We consider masked language modeling for long texts to be of particular importance, as it will allow finetuning for downstream tasks that need a context longer than the commonly used 512 tokens.

## Broader Impact

This work contributes towards the wider adoption of transformers by reducing their computational requirements; thus enabling their use on embedded or otherwise resource constrained devices. In addition, we have shown that for long sequences *clustered attention* can result to almost $50\%$ reduction in GPU training time which translates to equal reduction in CO2 emmisions and energy consumption.

## Acknowledgements

Apoorv Vyas was supported by the Swiss National Science Foundation under grant number FNS-30213 "SHISSM". Angelos Katharopoulos was supported by the Swiss National Science Foundation under grant numbers FNS-30209 "ISUL" and FNS-30224 "CORTI".

## Footnotes

*Work done at Idiap

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
