[Supplementary Material]

# Fast Transformers with Clustered Attention
# Supplementary Material

**Apoorv Vyas**[1,2]    **Angelos Katharopoulos**[1,2]    **François Fleuret**[2,3] [*]
[1]Idiap Research Institute, Switzerland
[2]Ecole Polytechnique Fédérale de Lausanne, Switzerland
[3]University of Geneva, Switzerland
`firstname.lastname@idiap.ch`

## 1   Scaling Attention with Fast Clustering

In this section we present graphical illustrations for the proposed *clustered* and *i-clustered* attention models in § 1.1 and § 1.2 respectively.

### 1.1   Clustered attention

In figure 1, we present the steps involved in *clustered* attention computation for an example sequence with $8$ queries and the number of clusters set to $3$. We first cluster the queries $Q$ using the K-means clustering to output $S$ which indicates the membership of queries to different clusters. We use different colors to represent different clusters. After clustering, the centroids $Q^c$ are used to compute the attention weights $A^c$ and the new values $V^c$ for the centroids. Finally, the values are broadcasted to get the new values $\hat{V}$ corresponding to each query.

Figure 1: Flow-chart demonstrating the compuation for *clustered* attention. We use different colors to represent the query groups and the computed centroids. The same colors are then used to show the attention weights $A^c$, new values for the centroids $\hat{V}^c$, and the resulting values $\hat{V}$ after broadcasting. For more details refer to § 1.1 or § 3.2 in the main paper.

---

[*]Work done at Idiap

## 1.2 Improved clustered attention

In this section, we first describe how we can efficiently compute the *i-clustered* attention using sparse dot products with the top-$k$ keys and values. We then present the flow chart demonstrating the same.

As discussed in the § 3.3 of the main paper, the improved attention matrix approximation $A_i^t$ for the query, $Q_i$ belonging to the cluster $j$ is computed as follows:

$$A_{il}^t = \begin{cases} \dfrac{\hat{m}_j \exp\left(Q_i K_l^T\right)}{\sum_{r=1}^N T_{jr} \exp(Q_i K_r^T)} & \text{if } T_{jl} = 1 \\ A_{il}^c & \text{otherwise} \end{cases}, \tag{1}$$

where, $T \in \{0, 1\}^{C \times N}$, stores the top-$k$ keys for each cluster. $T_{ji} = 1$ if the $i$-th key is among the top-$k$ keys for the $j$-th cluster and 0 otherwise.

As described in the main paper, $\hat{m}_j$ is the total probability mass on the top-$k$ keys for the $j$-th cluster given by:

$$\hat{m}_j = \sum_{r=1}^N T_{jr} A_{jr}^c. \tag{2}$$

Note that we can compute the attention weights $A_i^t$ on the top-$k$ keys by first taking sparse dot-product of $Q_i$ with the top-$k$ keys followed by the softmax activation and rescaling with total probablity mass $m_j$. For the rest of the keys, the attention weight is the clustered-attention weight $A_i^c$.

Similarly, the new values $\hat{V}_i$ can be decomposed into the following two terms,

$$\hat{V}_i = \hat{V}_i^t + \hat{V}_i^b, \tag{3}$$

where $\hat{V}_i^t$ is weighted average of the values corresponding to the top-$k$ keys with weights being the improved attention on the top-$k$ keys. $\hat{V}_i^b$ is the weighted average of the rest of the values with weights being the clustered attention $A_i^c$. The following equations show how we compute $\hat{V}_i^t$ and $\hat{V}_i^b$,

$$\hat{V}_i^t = \sum_{l=1}^N T_{jl} A_{il}^t V_l, \tag{4}$$

$$\hat{V}_i^b = \sum_{l=1}^N (1 - T_{jl}) A_{il}^c V_l, \tag{5}$$

Note that $\hat{V}_i^t$ is weighted average of $k$ values for each query and thus requires $\mathcal{O}\left(NkD_v\right)$ operations. $\hat{V}_i^b$ only needs to be computed once per-cluster centroid and thus requires $\mathcal{O}\left(NCD_v\right)$ operations.

In figure 2 we present the *i-clustered* attention computation for the same example sequence with 8 queries and the number of clusters and top-$k$ keys set to 3. The lower half of the figure shows the new value $\hat{V}^t$ computed by first taking sparse dot-products with the top 3 keys to get the attention weights. This is followed by taking the weighted average of the 3 correponding values. The top half of the figure shows the $\hat{V}^b$ computation. This is same as clustered attention computation but with attention weights corresponding to top 3 keys set to 0 for $A^c$. The resulting values $\hat{V}$ is the sum of $\hat{V}^b$ and $\hat{V}^t$.

## 2   Quality of the approximation

**Proposition 1.** *For the i-th query belonging to the j-th cluster, the improved clustered attention $A_i^t$ and clustered attention $A_j^c$ relate to the full attention $A_i$ as follows,*

$$\left\|A_i^t - A_i\right\|_1 \le \left\|A_j^c - A_i\right\|_1 \tag{6}$$

*Proof.* As discussed before, the improved attention matrix approximation $A_i^t$ for the query, $Q_i$ is computed as follows:

$$A_{il}^t = \begin{cases} \dfrac{\hat{m}_j \exp\left(Q_i K_l^T\right)}{\sum_{r=1}^N T_{jr} \exp(Q_i K_r^T)} & \text{if } T_{jl} = 1 \\ A_{il}^c & \text{otherwise} \end{cases}, \tag{7}$$

Figure 2: Flow-chart demonstrating the compuation for *i-clustered* attention. The lower half of the figure shows the new value $\hat{V}^t$ computed by sparse dot-products with the keys $K$ and values $V$ corresponding to the the top-$k$ keys in $T$. The top half of the figure shows the computation for $\hat{V}^b$ which is the weighted average of the rest of the values with weights coming from the clustered attention $A^c$. The resulting values $\hat{V}$ is the sum of $\hat{V}^b$ and $\hat{V}^t$. For more details refer § 1.2 or to the § 3.3 in the main paper.

where, $T \in \{0,1\}^{C \times N}$, stores the top-$k$ keys for each cluster, $T_{ji} = 1$ if the $i$-th key is among the top-$k$ keys for the $j$-th cluster and 0 otherwise. $\hat{m}_j$ is the total probability mass on the top-$k$ keys for the $j$-th cluster, computed as follows:

$$\hat{m}_j = \sum_{r=1}^{N} T_{jr} A^c_{jr}. \tag{8}$$

Given the full attention $A_i$, equation 7 can be simplified to

$$A^t_{il} = \begin{cases} \frac{\hat{m}_j}{m_i} A_{il} & \text{if } T_{jl} = 1 \\ A^c_{il} & \text{otherwise} \end{cases}, \tag{9}$$

where, $m_i$ is the total probability mass on the same top-$k$ keys for the $i$-th query, computed using the true attention $A_i$, as follows:

$$m_i = \frac{\sum_{r=1}^{N} T_{jr} \exp\left(Q_i K_r^T\right)}{\sum_{r=1}^{N} \exp\left(Q_i K_r^T\right)} \tag{10}$$

$$= \sum_{r=1}^{N} T_{jr} A_{ir}. \tag{11}$$

Without loss of generality, let us assume, $T_{jl} = 1 \quad \forall \quad l \in \{1, \ldots, k\}$ and $T_{jl} = 0 \quad \forall \quad l \in \{k+1, \ldots, N\}$.

In this case, equation 9 can be written as:

$$A^t_{il} = \begin{cases} \frac{\hat{m}_j}{m_i} A_{il} & \text{if } l \leq k \\ A^c_{il} & \text{if } l \geq k+1 \end{cases}. \tag{12}$$

The total probability masses on the top-$k$ keys, $m_i$ and $\hat{m}_j$ can now be expressed as:

$$m_i = \sum_{r=1}^{k} A_{ir}. \tag{13}$$

$$\hat{m}_j = \sum_{r=1}^{k} A_{jr}^c. \tag{14}$$

From equation 12 it is clear that the clustered attention, $A_i^c$, and the improved clustered attention, $A_i^t$, only differ on the keys $\{1, \ldots, k\}$. Thus, it suffices to show that $A_i^t$ has lower approximation error on these keys. The approximation error on the top-$k$ keys $\{1, \ldots, k\}$, let it be $e_t$, between the *i-clustered* attention and the *full* attention is as follows:

$$e_t = \sum_{l=1}^{k} \left| A_{il} - A_{il}^t \right| \tag{15}$$

$$= \sum_{l=1}^{k} \left| A_{il} - A_{il} \frac{\hat{m}_j}{m_i} \right| \tag{16}$$

$$= \sum_{l=1}^{k} A_{il} \left| 1 - \frac{\hat{m}_j}{m_i} \right| \tag{17}$$

$$= \left| 1 - \frac{\hat{m}_j}{m_i} \right| \sum_{l=1}^{k} A_{il} \tag{18}$$

$$= m_i \left| 1 - \frac{\hat{m}_j}{m_i} \right| \tag{19}$$

$$= |m_i - \hat{m}_j| \tag{20}$$

$$= \left| \sum_{l=1}^{k} A_{il} - A_{jl}^c \right| \tag{21}$$

$$\leq \sum_{l=1}^{k} \left| A_{il} - A_{jl}^c \right| \tag{22}$$

Therefore,

$$\left\| A_i - A_i^t \right\|_1 = \sum_{l=1}^{k} \left| A_{il} - A_{il}^t \right| + \sum_{l=k+1}^{N} \left| A_{il} - A_{il}^t \right| \tag{23}$$

$$= \sum_{l=1}^{k} \left| A_{il} - A_{il}^t \right| + \sum_{l=k+1}^{N} \left| A_{il} - A_{jl}^c \right| \tag{24}$$

$$\leq \sum_{l=1}^{k} \left| A_{il} - A_{jl}^c \right| + \sum_{l=k+1}^{N} \left| A_{il} - A_{jl}^c \right| \tag{25}$$

$$\leq \left\| A_i - A_i^c \right\|_1 \tag{26}$$

$$\square$$

## 3 Experiments

### 3.1 Time and Memory Benchmark

To measure the computational cost, we compare the memory consumption and computation time on artificially generated sequences of various lengths. For clustered attention we use 100 clusters, 63

bits for the LSH, and 10 Lloyd iterations for the K-Means. For the improved clustered attention, we use the same configuration with $k = 32$. For Reformer, we evaluate on two variants using 1 and 4 rounds of hashing. All models consist of 1 layer with 6 attention heads, embedding dimension of 64 for each head, and a feed-forward dimension of 1536.

In this experiment, we measure the required memory and GPU time *per single sequence element* to perform a forward/backward pass for the various self-attention models. Figure 3 illustrates how these metrics evolve as the sequence length increases from $N = 2^9$ to $N = 2^{15}$. For a fair comparison, we use the maximum possible batch size for each method and we divide the computational cost and memory with the number of samples in each batch and the sequence length.

We note that, in contrast to all other methods, vanilla transformer scales quadratically with respect to the sequence length and does not fit in GPU memory for sequences longer than $2^{13}$ elements. All other methods scale linearly. Clustered attention becomes faster than the vanilla transformer for sequences with 1000 elements or more, while improved clustered attention surpasses it for sequences with 2000 elements. Note that with respect to per sample memory, both clustered and improved clustered attention perform better than all other methods. This can be explained by the fact that our method does not require storing intermediate results to compute the gradients from multiple hashing rounds as Reformer does. It can be seen, that lsh-1 is faster than the improved clustered clustered attention, however, as also mentioned by [1] Reformer requires multiple hashing rounds to generalize.

(a) Per Element Time        (b) Per Element Memory

Figure 3: Per element GPU time and memory consumption for a forward/backward pass. All models, except full, scale linearly with respect to the sequence length since they have constant time and memory per element. Detailed analysis can be found in § 3.1.

## 3.2   Ablation on clusters and sequence length

Following [1], we introduce a synthetic task to analyze the relationship between the number of clusters and sequence length. In our task, the transformer models need to copy some symbols that are masked out from either the first or second half of the sequence. In particular, we generate a random sequence of tokens and we prepend a unique separator token, let it be 0. The sequence is then copied to get a target of the form $0w0w$, where $w \in \{1, \ldots, C\}^L$, $C$ is the number of possible symbols and $L$ is the sequence length. To generate the input, we replace some symbols from the first half of the sequence and some different symbols from the second half, such that the target sequence can be reconstructed from the input. An example of an input output pair with $L = 4$ can be seen in figure 5. Note that to solve this task, transformers simply need to learn to attend to the corresponding tokens in the two identical halves of the sequence.

| Input | 0 | 4 | M | 2 | 2 | 0 | 4 | 5 | M | 2 |
|---|---|---|---|---|---|---|---|---|---|---|
| Output | 0 | 4 | 5 | 2 | 2 | 0 | 4 | 5 | 2 | 2 |

Figure 5: Example of an input and output pair for the masked copy task. M denotes the masked out tokens.

**Accuracy with respect to clusters and hashing rounds**

| (a) Improved clustered | (b) Clustered | (c) Reformer |

Figure 4: The heatmaps depict the achieved accuracy on an artificial copy task (§ 3.2) as the sequence length, the number of clusters and the number of hashing rounds varies. Improved clustered (4a) is the only fast transformer variant that can solve the task perfectly for any sequence length and number of clusters combination.

We set the sequence length $L$ to one of $\{31, 63, 127, 255\}$ which means the input length varies between $N = 2^6$ and $N = 2^9$. For each sequence, we sample tokens uniformly from $\{1, \ldots, 10\}$ and randomly mask out $20\%$ of the tokens. To analyze the impact of number of clusters on performance, we train full transformer as well as clustered variants with different number of clusters and Reformer with different number of hashing rounds.

All transformer variants consist of 4 layers, 6 attention heads, embedding dimension of 32 for each head, and feed-forward dimension of 768. For both clustered and improved clustered attention, we set the number of bits for LSH to 63 and the number of Lloyd iterations for the K-Means to 10. Both clustered and improved clustered attention are trained with 15, 30, 60 and 100 clusters. We also train Reformer with 1, 4, 8 and 16 hashing rounds. Finally, all models are trained using R-Adam optimizer [2] with a learning rate of 0.0002, batch size of 32 for 5000 iterations.

In figure 4, we illustrate the results of this experiment as heatmaps depicting the achieved accuracy for a given combination of number of clusters and sequence length for clustered transformers and number of hashing rounds and sequence length for Reformer. Note that the vanilla transformer solves the task perfectly for all sequence lengths. We observe that both clustered (Fig. 4b) and Reformer (Fig. 4c) require more clusters or more rounds as the sequence length increases. However, improved clustered achieves the same performance as vanilla transformers, namely *perfect accuracy*, for every number of clusters and sequence length combination. This result increases our confidence that the required number of clusters for our method is not a function of the sequence length but of the task at hand.

## 3.3 Automatic Speech Recognition

In this section, we present the details for the ASR experiments such as transformer architecture, optimizer and learning rate schedule. As mentioned in the main paper, for *i-clustered*, unless specified, $k$ is set to 32. Furthermore, all transformers have 6 heads with an embedding dimension of 32 on each head and feed-forward dimension of 768. Other architectural details specific to experiments are described later.

### 3.3.1 Wall Street Journal

**Convergence Behaviour:**

For this experiment, we train transformer with full, clustered and Reformer attention variants. All models consist of 9 layers. For Reformer, we train two variants with 1 and 4 rounds of hashing with chunk size fixed to 32 as suggested. For clustered and improved clustered attention we set the number of clusters to 100. We also set the number of Lloyd iterations for K-Means to 10 and the bits for LSH

to 63. All models are trained to convergence using the R-Adam optimizer [2] with a learning rate of 0.0001, max gradient norm set to 10.0 and and weight decay of 0.01. The learning rate is dropped when the validation loss plateaus. For each model we select the largest batch size that fits the GPU. The *full* attention model was trained with a batch size of 2 while the clustered variants: *clustered* and *i-clustered* could fit batch sizes of 14 and 10 respectively. For Reformer variants: *lsh*-1 and *lsh*-4, batch sizes of 8 and 6 were used.

(a) Wall Street Journal        (b) Switchboard

Figure 6: We show training/validation loss convergence for different transformer variants. Only *i-clustered* has a faster or comparable wall-clock convergence to full attention. Both the clustered variants are have a significantly better convergence than both *lsh*-1 and *lsh*-4. Note that due to a smaller batch size *full* makes many more updates than all other transformer variants. More details can be found in § 3.3.1 and § 3.3.2.

In figure 6a, we show the training loss convergence for different transformer variants. It can be seen that *i-clustered* has a much faster convergence than the *clustered* attention. This shows that the improved clustered attention indeed approximates the full attention better. More importantly, only the *i-clustered* attention has a comparable wall-clock convergence. Given that *full* has a much smaller batch size, it makes many more updates per-epoch. We think that a slightly smaller batchsize with more updates would have been a better choice for the clustered transformers w.r.t. the wall-clock convergence. This is reflected in the Switchboard experiments where the batchsizes for clustered variants were smaller due to more layers. Finally, as can be seen from the wall-clock convergence, the clustered transformers significantly outperform the Reformer variants.

**Speed-Accuracy Tradeoff:**

As described in the main paper, for this task we additionally train *full* with 4 and 6 layers. Similary, we train *clustered* with 9 layers, and 200 and 300 clusters. We also train an *i-clustered* model with 9 layer and 200 clusters, and smaller models with 6 layers, and 100 and 200 clusters.

For *clustered* and *i-clustered* variants with 9 layers, we finetuned the previously described models trained with 100 clusters. We finetuned for 15 epochs with a learning rate of 0.00001. We train *full* with 4 and 6 layers to convergence in a similar fashion to the *full* with 9 layers described previously. Finally, for *i-clustered*, we first trained model with 6 layers and 100 clusters using the training strategy used for 9 layers and 100 clusters. We then finetuned this model for 15 epochs using 200 clusters and a learning rate of 0.00001.

### 3.3.2 Switchboard

**Convergence Behaviour:**

For this experiment, we train transformer with full and clustered attention variants. All models consist of 12 layers. For clustered and improved clustered attention we set the number of clusters to 100. We also set the number of Lloyd iterations for K-Means to 10 and the bits for LSH to 63.

Following common practice for flat-start lattice-free MMI training, we train over multiple gpus with weight averaging for synchronization as described in [5]. Specfically, we modify the *e2e* training recipe for the **Wall Street Journal** in Kaldi [4] with the following two key differences: first, the acoustic model training is done in PyTorch using the PkWrap toolkit [3] and second, we use R-Adam optimizer instead on natural stochastic gradient descent.

All models are trained using the R-Adam optimizer with a learning rate of 0.0002, max gradient norm set to 10.0 and and weight decay of 0.01. The learning rate is dropped when the validation loss plateaus. We use the word error rate (WER) on the validation set for early stopping and model selection. The *full* attention model is trained with a batch size of 2 while the clustered variants: *clustered* and *i-clustered* are trained with a batch size of 6.

In figure 6b, we show the training loss convergence for different transformer variants. It can be seen that *i-clustered* has the fastest convergence for this setup. Note that the overall training time for *clustered* attention is still less than that of *full* as it starts to overfit early on the validation set WER.

**Speed-Accuracy Tradeoff:**

For this task we additionally train *full* with 6 and 8 layers. Similary, we train *clustered* with 12 layers, and 200 and 300 clusters. We also train *i-clustered* with 12 layer and 200 clusters, and smaller models with 8 layers, and 100 and 200 clusters.

For *clustered* and *i-clustered* variants with 12 layers, we finetuned the previously described models trained with 100 clusters. We finetuned for 5 epochs with a learning rate of 0.00001. Once again, *full* with 6 and 8 layers were trained to convergence similar to *full* with 12 layers described previously. Finally, for *i-clustered* with 8 layers, we first train a model with 100 clusters using the training strategy used for 12 layers and 100 clusters. We then finetuned this model for 5 epochs using 200 clusters and a learning rate of 0.00001.

### 3.4 RoBERTa Approximation

In this section we provide a qualitative comparison between the *full* attention, and the clustered attention variants *clustered* and *i-clustered* used for approximation. As described in main paper, we use 25 clusters for both attention variants. In Figure 7 we show the attention distribution for the question tokens for a randomly selected question-context tuple from the SQuAD dataset. For each token in the question we show the attention distribution over the input sequence formed by concatenating question and context tokens with *CLS* and *SEP* tokens appended. It can be seen that with only few clusters, improved clustered approximates the full attention very closely even when the attention distribution has complicated and sparse patterns. In contrast, clustered attention fails to capture such attention distribution during approximation. Moreover, it can further be seen that for almost all question tokens, both full and improved clustered have the same tokens with the highest attention weights. This further strengthens our believe that improved clustered attention can approximate a wide range of complicated attention patterns.

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

Manning finished the year with a career-low 67.9 passer rating, throwing for 2,249 yards and nine touchdowns, with 17 interceptions. In contrast, Osweiler threw for 1,967 yards, 10 touchdowns and six interceptions for a rating of 86.4. Veteran receiver Demaryius Thomas led the team with 105 receptions for 1,304 yards and six touchdowns, while Emmanuel Sanders caught 76 passes for 1,135 yards and six scores, while adding another 106 yards returning punts. Tight end Owen Daniels was also a big element of the passing game with 46 receptions for 517 yards. Running back C. J. Anderson was the team's leading rusher 863 yards and seven touchdowns, while also catching 25 passes for 183 yards. Running back Ronnie Hillman also made a big impact with 720 yards, five touchdowns, 24 receptions, and a 4.7 yards per carry average. Overall, the offense ranked 19th in scoring with 355 points and did not have any Pro Bowl selections.

(a) *context*

(b) *full*

(c) *improved-clustered*

(d) *clustered*

Figure 7: Attention matrices for question-context tuples for *full* attention, and *clustered* and *i-clustered* attention used for approximation. 7a shows the the context for the question with answer higlighted in red. 7b shows the attention distribtution for *full*, 7c and 7d show the approximation using *i-clustered* and *clustered* respectively. Note that *i-clustered* has attention patterns very similar to *full* while *clustered* shows qualitatively different attention patterns. For each question token, we also present the tokens with highest attention above a threshold on the right axis. For more information refer to § 3.4.