[Reviews · NeurIPS 2020]

Review 1

Summary and Contributions: The paper proposes the clustered attention which aims at reducing the computational cost of self-attention, by clustering the queries and run attention using a single representative vector from every cluster on behalf of all the queries in that cluster and broadcast the output values to the individual queries based on their cluster membership. Given a fixed number of clusters, this leads to linear complexity as opposed to the quadratic complexity of the vanilla self-attention. -------------------------------- Authors submitted a response for the rebuttal. I appreciate them taking their time to submit responses to points raised by the reviewers and make things clear. As for the responses, I would like to ask authors to add this point to the camera-ready version of the paper that how cluster attention (the basic version) can have different clustering schemes across different layers.

Strengths: The paper is really well written and it motivates the problem really good. The idea is presented sound and clear and there are some theoretical discussions on how far the approximation of the attention distribution can be from the distribution computed by a standard self-attention (i.e. with O(N^2)). The experimental setup seems correct and the results are to some extent supporting of the claims made in the paper.

Weaknesses: There are a few points that are not clear from the paper, which I list below: - As far as I understood in the clustered attention (not the improved one), the value of the i-th query becomes the value of the centroid of the cluster that the query belongs to. So after one round of applying the clustered attention, we have a set C distinct values in N nodes. I wonder what is the implication of this for the next round of the clustered attention, because there is no way to have two nodes that were in the same cluster in the previous round to be in different clusters in the next round (as their values will be the same after round 1) and the only change in the clustering that makes sense is merging clusters (which is not the case as apparently the number of clusters stays the same). Isn’t this too restrictive? What if the initial clustering is not good, then the model has no chance to recover? If the number of clusters stays the same, does the clustering in the layer after layer 1 does anything different than the clustering in the layer 1 (if not they're removable)? - It’s a bit unclear if LSH-X is the Reformer, or a simpler version of the reformer (LSH Transformer). The authors mentioned that the Reformer can’t be used in a setup with heterogeneous queries and keys. First of all, I think it shouldn't be that hard to modify Reformer to support this case. Besides, authors don’t have any task in that setup to see how well the clustered attention does when the clustered queries are not the projections of the inputs that the keys are projected from. - The experiments that are done in the setup that the model has to deal with long sequences is limited to a single modality. Would be nice to have the model evaluated on large inputs in vision/text/algorithmic tasks as well. - Although the method is presented nicely and the experiments are rather good and complete, a bit of analysis on what the model does, which can be extremely interesting, is missing (check the feedback/suggestions). - The authors only consider vanilla transformer and (I think an incomplete version of) Reformer, while there are obvious baselines, e.g. Longformer, sparse transformer, or even Local attention (check the feedback/suggestions).

Correctness: The claim is to reduce the complexity of the attention to the linear time by approximating the attention distribution via group-level querying. The method seems correct and the theoretical discussion supports these claims.

Clarity: The paper is well written. The motivation, the background section, and the methodology look great. I also really appreciate the diagrams in the appendix.

Relation to Prior Work: Authors positioned their work really well in the existing related research and in case of super close baselines, they explained how their method is different from that baseline. However, this comparison is pretty limited in the experiments and empirical section, and most of the baselines are simply ablated versions of the proposed model. Just as another related work, the idea presented in this paper reminded me of Set Transformer paper, where they use a similar idea for pooling and as queries, they learn a set of $C$ seed vectors, where $C << N$, as opposed to what is done here, where seed vectors are not learned, but depends on the inputs and are in fact centroid of clusters. Also in the idea proposed here, after computing $C$ values (which is a pooled version of input), it broadcasts it back to $N$ values, by taking the cluster memberships into account.

Reproducibility: Yes

Additional Feedback: I have a few suggestions to improve the paper: Adding a simple baseline: group nodes simply based on their position into fixed-size groups (those next to each other go to the same group) and simply create a group query by averaging the queries of nodes in that group and then do the same thing as the clustered attention. (and for instance, in the next layer, shift the boundaries to have the chance of getting different nodes grouped together!) Check what nodes are clustered into the same group. Is the clustering criterium dominated by proximity or content, or none? To me, making a transformer efficient is directly related to applying transformer on vision as the inputs are high dimensions (even on MNIST, the length of the input-with no factorization- is 784). So it makes it super exciting to see how clustered attention does in for instance CIFAR10/ImageNet classification/generation. Also, it gives the chance of visually inspecting the clusters and do cool analysis.


Review 2

Summary and Contributions: The paper proposed an efficient approximation of the Transformer model which could efficiently reduce the computation of self-attention to linear complexity. The key is to cluster the queries into a fixed number of groups, while during the attention mechanism, instead of using a new query each time, use the fixed number of centroids of the clusters.

Strengths: Surprisingly this change doesn’t hurt the performance, and achieves significant speedup. Linear performance is theoretically better than Reformer, and experimentally it has proved that it is significantly faster and better in practical ASR tasks. The paper is also sound in its proofs. They provide convincing bounds for the approximation of using cluster centroids.

Weaknesses: One caveat could be, the authors are applying their proposed method on speech data, where the queries tend to be similar within frames close to each other. It’d be more convincing to see this method to be applied on an NLP task, such as language modeling and machine translation.

Correctness: Yes.

Clarity: Yes.

Relation to Prior Work: Yes. The paper has clearly discussed this point in different kinds transformer modifications. There are other related works that the authors might want to investigate: 1. Pay-less-attention (https://arxiv.org/pdf/1901.10430.pdf), which reduces to linear complexity for the self-attention part of the model, by using convolutions. 2. Longformer (https://arxiv.org/pdf/2004.05150.pdf), Using dilated convolution plus block-wise attention to reduce complexity.

Reproducibility: Yes

Additional Feedback:


Review 3

Summary and Contributions: The paper proposes an efficient way of computing self-attention by clustering queries and then only using their centroids in attention computation. The quality of this approximation can be improved by re-computing the attention between the top-K keys and actual queries. For efficiency, a hashing followed by k-means clustering is used. The method is evaluated on ASR tasks, and ROBERTA approximation.

Strengths: The paper tackles an important problem of improving the efficiency of self-attention, which has a large practical impact as Transformers are being used everywhere. The proposed idea is quite simple and intuitive. The idea of clustering queries is itself not novel as it's used in Reformer, but using their centroid to find top-K keys is a novel technique. Unlike Reformer, the proposed method seem simpler to implement and does work well with pre-trained models without any fine-tuning.

Weaknesses: I have some concerns about the actual efficiency of the method in practice. Usually, attention weights are computed by matrix multiplication as all queries attend to all keys. But in the improved clustering, some queries would attend to some keys. Although it is theoretically has less FLOPs, it is not clear if it's actually fast on GPU to compute this block sparse (as mentioned in L194) matrix multiplication? In the experiments, the number of clusters needed for a good performance is not significantly smaller than the actual input size (200 vs 780, 200 vs 534) as claimed in L130. If C has to grow with N, then the method is still quadratic in its complexity. Further, the actual training time is only slightly faster as shown table 2 and has a worse performance than the baseline. Therefore, I have some doubt if the proposed method is a practical way of making Transformers faster.

Correctness: Yes. I only have minor comments: - In the experiments, only the forward time is reported. It's not clear how this time is computed. Is it during training or training? Also, is this measured with batches or a single sample at a time. The important metric is the batched training time since it will determine training time.

Clarity: Yes, the paper is written clear and concise. I have few minor comments: - L77: Transformer-XL is described as "recurrent", which is not true as it's strictly a feed-forward model. - L83: those methods do not have a quadratic complexity with regards to the sequence length. Attending previous M steps for N steps has a complexity of O(MN), and it can be M << N. - L163: Is the clustering complexity O(NCD) so bad? The algorithm already has that complexity as show in L143. - What is the actual speed improvement on ReBERTa?

Relation to Prior Work: Yes, it's clear how this work differs from other methods.

Reproducibility: Yes

Additional Feedback: == Post-rebuttal comment == I will keep my score as it is.

[Author Response · NeurIPS 2020]

We would like to thank the reviewers for their helpful comments and their thorough evaluation of our work. We are encouraged that reviewers find our paper clear and well written (R1, R2, R3) and our method to be theoretically sound (R1, R2), of high practical impact (R2, R3), and intuitive and easy to implement (R3). Below, we address the issues that have been raised.

**[R1] Should the clusters merge in the next layer?**: The output of the attention layer involves (a) residual connections: this means queries in the same cluster can have different values (b) multi-head attention: this allows for a query to be clustered differently in two different heads. This is followed by a feed-forward projection, which results in new clustering patterns in the subsequent layers.

**[R1] Is LSH-X a simpler version of the Reformer?**: LSH-X refers to Reformer as described in the paper except for the reversible layers. Reversible layers is a technique introduced by Gomez et al. (2017) and is orthogonal and independent to both clustered and LSH attention. As our focus is on the efficiency of the attention implementation, we do not consider the reversible layers for both methods.

**[R1] Can Reformer be used for heterogeneous queries and keys?**: Using Reformer with heterogeneous queries and keys, requires significant changes in its core component, namely sorting and chunking the hash values to create query/key groups. In contrast, clustered attention places no such restriction.

**[R1] Analysis of clusters and Set Transformers**: Thank you for the suggestions, we will add further analysis of the clustering in the supplementary material as suggested. We will also add Set Transformers to the related work section.

**[R1, R2] Experiments on other modalities. Is speech favorable to clustering?**: Thank you for your comments. We would like to mention that our NLP approximation experiment for GLUE and SQuAD tasks in § 4.3 shows that clustered attention can capture arbitrarily complex attention patterns. In the future, we will also explore applications to NLP/vision tasks in the long context setting, as suggested.

**[R1, R2] Other baselines to investigate: Longformer, Sparse Transformer, Pay-less-attention, and Local attention**: We thank R1, R2 for the valuable suggestions. Sparse Transformer's reliance on the blocksparse library and the lack of documentation significantly hinder its evaluation on new tasks (evident by the lack of comparisons in the community). Longformer is a very recent work that we will compare with in the future. Upon R1's suggestion, we evaluate local attention on the synthetic copy task (§ 3.2 in the supplementary). We observe that for a sequence of length 128, local attention with context of 32 and 4 layers cannot solve the task whereas all clustered attention variants can solve it perfectly with 30 clusters.

**[R3] Is sparse dot product used in improved-clustered attention practically efficient?**: We refer R3 to Fig. 3 of the supplementary, where we present the time-memory benchmark. Notably, improved attention is faster than vanilla for sequence lengths greater than 1024. Unlike general sparse dot products, our implementation exploits the fact that all queries in a cluster use the same top-k keys. This allows us to cache the keys in CUDA shared memory for better performance. We will add details about our implementation in the supplementary material.

**[R3] If C grows with N, then the method has quadratic complexity**: We agree with R3 that C should be small for good performance. Note that, C is not necessarily a function of N as it depends heavily on the task. We would like to refer R3 to an ablation on the relationship between clusters and sequence length in § 3.2 of the supplementary. For the masked copy task, improved-clustered solves the task for every sequence length (up to 512) with just 15 clusters. Similarly, for SQuAD approximation, using only 25 clusters for a sequence length of 380 leads to good performance.

**[R3] Training time (table 2) is only slightly faster with worse performance than the baseline?**: While for CTC loss (table 2), the training time improvements were only small, for LF-MMI loss (table 3), the improvements were significant. Also note that while the average sequence lengths for WSJ and Switchboard are 780 and 530, the maximum lengths are 2500 and 3850 respectively. Finally, during inference under fixed computational budget, improved-clustered is consistently better than the baselines (Fig. 1).

**[R3] Is clustering complexity O(NCD) so bad? What is actual speed improvement on RoBERTa? Minor comments**: Thank you for the suggestion, we agree that for long sequences, k-means with euclidean distance O(NCD) could improve performance due to better clustering. For RoBERTa approximation, vanilla attention is faster due to small sequences. The purpose of the experiment is to understand the ability of the clustered attention variants to capture complex attention distributions using only a few clusters. We will make it clear in the camera-ready version that vanilla attention is faster in this experiment. We also want to thank R3 for pointing out the mistakes, we will fix them.

[Meta-Review · NeurIPS 2020]

Three reviewers recommended to accept the paper. The reviewers liked the idea introduced in the paper, which tackles an important research question (making self attention more efficient when applied to long sequences). Minor concerns included missing previous works (from discussion and experiments), and the fact that it is not clear if the method would lead to important gains in practice (as number of clusters might not scale sublinearly, and the work relies on sparse matrix multiplication needing efficient GPU implementation). After rebuttal and discussion, the reviewers indicated that these concern were addressed or did not justify a rejection. Thus, the paper is accepted. I encourage the authors to take into account the feedback of the reviewers, and to cite missing existing work, including "Efficient content-based sparse attention with routing transformers" (https://arxiv.org/abs/2003.05997).